

# Graph neural networks embedded with domain knowledge for cyber threat intelligence entity and relationship mining

Gan Liu[1], Kai Lu[1,2] and Saiqi Pi[3]

[1] School of Cyberspace Security (School of Cryptology), Hainan University, Haikou, China
[2] Department of Public Safety Technology, Hainan Vocational College of Politics and Law, Haikou, China
[3] College of Big Data and Information Engineering, GuiYang Institute of Humanities and Technology, Guiyang, China

## ABSTRACT

The escalating frequency and severity of cyber-attacks have presented formidable challenges to the safeguarding of cyberspace. Named Entity Recognition (NER) technology is utilized for the rapid identification of threat entities and their relationships within cyber threat intelligence, enabling security researchers to be promptly informed of the occurrence of cyber threats, thereby enhancing the efficiency of security defense and analysis. However, current models for identifying network threat entities and extracting relationships suffer from limitations such as the inadequate representation of textual semantic information, insufficient granularity in threat entity recognition, and errors in relationship extraction propagation. To address these issues, this article proposes a novel model for Network Threat Entity Recognition and Relationship Extraction (CtiErRe). Additionally, it redefines seven network threat entities and two types of relationships between threat entities. Specifically, first, domain knowledge is collected to build a domain knowledge graph, which is then embedded using graph convolutional networks (GCN) to enhance the feature representation of threat intelligence text. Next, the features from domain knowledge graph embedding and those generated by the bidirectional encoder representations from transformers (BERT) model are fused using the Layernorm algorithm. Finally, the fused features are processed using the GlobalPointer algorithm to generate both the threat entity type matrix and the threat entity relation type matrix, thereby enabling the identification of threat entities and their relationships. To validate our proposed model, we conducted extensive experiments, and the results demonstrate its superiority over existing models. Our model performs remarkably in threat entity recognition tasks, with accuracy and F1 scores reaching 92.13% and 93.11%, respectively. In the relationship extraction task, our model achieves accuracy and F1 scores of 91.45% and 92.45%, respectively.

Corresponding author
Saiqi Pi, saiqipi@foxmail.com

# INTRODUCTION

With the rapid development of Internet technology, various information systems are widely used in people's lives, thereby making social development more efficient and orderly. However, this high degree of informatization has also brought many security risks in the field of cyberspace, leading to an increasingly serious situation in network security (*Siddiqui, Yadav & Husain, 2018*). According to statistics, more than 23,000 novel instances of malware and 20 fresh vulnerabilities are disseminated on a daily basis, with the occurrence of approximately 110,000 cyber assaults each hour (*Samtani et al., 2020*). To defend against network attacks and eliminate network security risks, traditional network security protection methods often rely on deploying network security tools at critical points within the organization or at network boundaries. These security tools include firewalls (*Wu et al., 2023*), intrusion detection systems (IDS) (*Thakkar & Lohiya, 2023*) and intrusion prevention systems (IPS) (*Kumar et al., 2022*). Such security protection systems primarily execute network security static control policies based on feature detection and predefined rule matching, monitoring network threats from multiple dimensions. The content of detection and monitoring mainly includes viruses, malware, vulnerabilities, malware, and traffic (*Kumar et al., 2022*). However, with the rapid development of emerging fields such as cloud computing, big data, and artificial intelligence, security threats are evolving towards generalization and complexity. As network attacks become increasingly creative, exhibiting greater concealment and persistence, they present more challenges to network security defense (*Sun et al., 2022*). In response to these evolving threats, security researchers have collected and analyzed a vast array of network threat information to discern current and emerging attack trends, an initiative known as Cyber Threat Intelligence (CTI) (*Wagner et al., 2019*; *Eltayeb, 2024*). By gathering multi-source CTI data, security professionals can gain a comprehensive view of the threat landscape and uncover threat detection indicators that might otherwise remain obscured in isolated sources. CTI refers to the collected and analyzed knowledge about network threats, encompassing attackers' motivations, objectives, and methods. Business professionals can utilize this curated knowledge of network threats to safeguard their organization's core assets against potential attacks (*Sun et al., 2023*). Constructing cyber threat intelligence involves a wide spectrum of knowledge related to network security threats, including attack behaviors, threat actors, targets, attack tools, malware, and vulnerabilities (*Barnum, 2012*). To swiftly comprehend the ever-evolving landscape of network threats and shield against complex, persistent, organized, and weaponized cyberattacks, organizations worldwide are increasingly inclined to engage in information sharing *via* CTI. Furthermore, there is a growing adoption of various CTI formats and standards. Structured Threat Information eXpression (STIX) (*Barnum, 2012*; *Sun et al., 2023*), Trusted Automated eXchange of Indicator Information (TAXII) (*Kokkonen, 2016*), Cyber Observable eXpression (Cybox) (*Qamar et al., 2017*; *Densham, 2015*), Malware Attribute Enumeration and Characterization (MAEC) (*Kirillov et al., 2011*; *Noor, Abbas & Shahid, 2018*), and other standards have been successively proposed. These standards facilitate the construction of threat intelligence reports from various levels of network

threats and threat indicators. However, in most cases, descriptions of network threats are published in natural language text, and transforming these texts into well-defined threat intelligence formats requires a significant amount of manual effort, which can be quite laborious. Automatically, accurately, and rapidly extracting threat behavioral entities from a massive volume of threat intelligence texts has become a focal point of attention in both academia and industry. Understanding complex behavioral relationships within network threat technical texts is recognized as a challenge in the field of natural language processing (NLP).

In order to obtain descriptive or static CTI data from unstructured texts, researchers have proposed a variety of information extraction methods (*Cook & Jensen, 2019*; *Liao et al., 2016*; *Zhu & Dumitras, 2018*; *Dong et al., 2019*; *Zhao et al., 2020*). While these methods have achieved promising results, they still face several challenges:

1) Existing methods lack an understanding of semantic knowledge specific to the field of cybersecurity in terms of word vector representations and pretraining models, and they also show suboptimal performance in recognizing nested threat entities.

2) The identification of threat behavioral entities primarily focuses on indicators of compromise (IOC), but threat intelligence encompasses various behaviors, with IOCs being just one part of it.

3) Most threat entity relation extraction methods in the field of cybersecurity use recursive and discriminative approaches to confirm relationships. However, due to the lack of domain knowledge about threat entities, these methods perform poorly in recognizing relationships between new threat entities.

To address the issues in threat entity extraction and relationship extraction mentioned above, we propose a new model for threat entity recognition and relationship extraction, named the CtiErRe model. The training process of the CtiErRe model is illustrated in Fig. 1, and it can be broadly divided into two modules: the data processing module and the model construction module. This model combines graph neural networks (GCN) (*Zhou et al., 2020*), bidirectional encoder representation from transformers (BERT) (*Devlin, 2018*), and the GlobalPointer algorithm (*Su et al., 2022*). Specifically, we first employ web crawling techniques to collect text of network threat entities. We compile a list of words related to threat behaviors through word frequency statistics. We calculate an overall word-domain knowledge matrix using the Pointwise Mutual Information (PMI) algorithm. This knowledge matrix is used to construct a domain knowledge graph for the text. Next, The GCN model is utilized to learn text features with structural information from the generated domain knowledge graph. Subsequently, the LayerNorm algorithm is employed to integrate the text features with structural information obtained from the GCN model with the features derived from the BERT model. Finally, we use the GlobalPointer algorithm to classify the fused features. This classification involves distinguishing between seven types of threat behavioral entities and two types of entity relationships. The CtiErRe model employs a GCN to extract structural information from threat-related texts. It utilizes a BERT model to capture textual features of the threats. Furthermore, it applies the

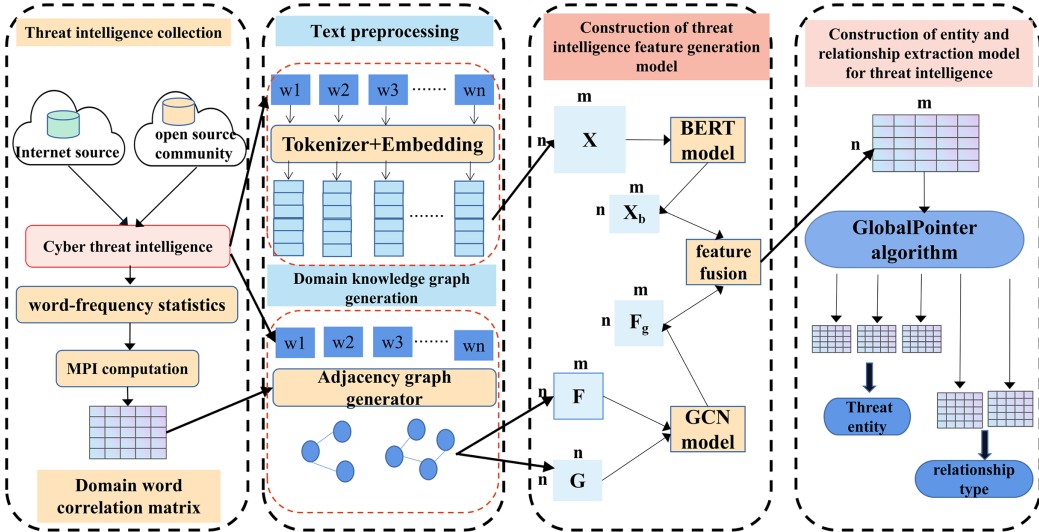

**Figure 1 Entity recognition and relationship extraction model of threat behavior.**

GlobalPointer algorithm for the ultimate identification of entities and relationships, thereby mitigating the propagation of errors and enhancing the precision of recognizing nested relationships. To broaden the scope of threat behavioral entities, we have delineated seven specific types: threat actor, intrusion set, attack pattern, tool, malware, vulnerability, and threat object. Additionally, we have defined the relationships between these entities, categorizing them into "uses" and "targets". These relationships are illustrated in Fig. 2. Concurrently, our model's performance has been verified, with an F1 score of 93.11% for entity recognition and 92.45% for relationship recognition.

## In summary, the main contributions of this study are as follows:

- A textual knowledge graph matrix within the domain of cyber threat intelligence was constructed, analyzing the semantic relationships between threat entities, thereby providing structural textual information for the tasks of threat entity recognition and relationship extraction.
- To enhance domain knowledge representation and improve the recognition of nested network threat entities, we designed the CtiErRe model. This model combines the GCN model, BERT model, and the GlobalPointer algorithm. The multi-model integration approach enables the extraction and analysis of threat entities and their relationships from different perspectives. The model is capable of simultaneously performing network threat entity extraction and relation extraction between entities.
- Due to the differences between the knowledge features extracted by the GCN model and the textual features extracted by the BERT model, a more effective feature fusion approach is needed. In this article, we propose a layer normalization fusion method, which normalizes features from different statistical domains into a common statistical scale.

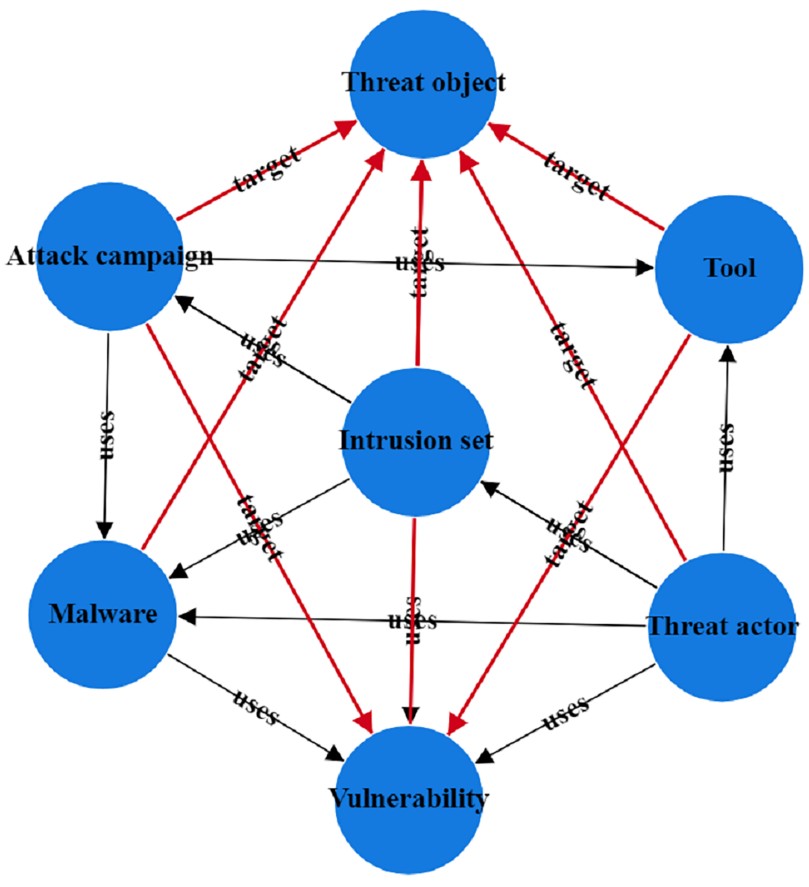

**Figure 2** **Entity relationship diagram of threatening behavior.**

# RELATED WORKS

## Research on network threat intelligence

The essence of cyber threat intelligence is a form of manifestation of big data in cybersecurity. It is not merely the collection and summarization of threat information, but an integrated data service that encompasses both intelligence sharing and analysis. To facilitate its rapid distribution and sharing, the industry established a series of related standards that make intelligence executable, *i.e.*, machine-readable. These standards include STIX (*Barnum, 2012*; *Sun et al., 2023*), TAXII (*Kokkonen, 2016*), Cybox (*Qamar et al., 2017*; *Densham, 2015*), and the Common Attack Pattern Enumeration and Classification (CAPEC) (*Riera et al., 2022*; *Kotenko & Doynikova, 2015*). Furthermore, to effectively utilize threat intelligence, it is necessary to employ diverse technological means to gather large-scale, fragmented threat-related information from various sources. This information should then be centrally deep-mined, refined, and integrated to form a collection of threat indicators closely linked to core information system assets. Based on the correlation analysis of threat elements, this collection can guide users in devising effective security response strategies.

STIX is a language and serialization format for exchanging cyber threat intelligence in cyberspace (*Barnum, 2012*; *Sun et al., 2023*). Defined and developed by The MITRE Corporation, STIX has evolved to version 2.0. The language aims to cover all aspects of threat information and make the expression of threat intelligence as comprehensive as possible. It also strives to ensure that threat intelligence data is flexible, scalable, automated, and interpretable. STIX consists of nine key components and is applicable in four different scenarios, employing XML for encoding. Adhering to STIX format standards allows for the comprehensive representation of all forms of cyber threat intelligence, enabling consistent sharing, storage, and analysis of threat intelligence data in a standardized manner.

TAXII is a standard for information exchange used to share cyber threat intelligence across product lines, services, and organizational boundaries (*Kokkonen, 2016*). It provides the transport mechanism for STIX's structured threat information and supports three modes: source/subscription, centralized, and peer-to-peer. TAXII offers four services: discovery, collection management, inbox, and information retrieval. Users can select and combine these services to meet their specific needs, creating composite services as needed.

CybOX is a method for representing computer observable objects and network dynamics and entities (*Qamar et al., 2017*; *Densham, 2015*). It was integrated into STIX 2.0. Observable objects were static assets or dynamic events. The CybOX specification provides a standardized and extensible syntax for describing all content that can be observed from a computing system and operation. It can be used for threat assessment, log management, malware feature description, indicator sharing, event response, and more.

CAPEC is a publicly available classification method dedicated to providing a common taxonomy of common attack patterns (*Riera et al., 2022*; *Kotenko & Doynikova, 2015*). Its purpose is to assist users in understanding how adversaries exploit vulnerabilities in applications and other network-supporting capabilities to carry out attacks. CAPEC currently encompasses over 500 attack types. The format of CAPEC is structurally similar to the CWE Vulnerability Classification Library and includes three classification tables and one reference table. The classification tables categorize information based on different dimensions.

## Research on entity identification of network threat intelligence

The cyber threat intelligence is mostly described in the form of natural language text and published on network platforms. This text-based description of network threats is unstructured. The current research focus is on how to quickly identify relevant threat behavioral entities from this unstructured text. *Liao et al. (2016)* proposed a method that uses NLP to automatically extract IOC from blog articles. *Husari et al. (2017)* introduced the TTPDrill algorithm, which utilizes NLP and information retrieval (IR) to extract threat actions from unstructured CTI text. *Gao et al. (2020)* designed a threat intelligence metamodel to describe the semantic relevance of infrastructure nodes. They trained a network threat intelligence model using a heterogeneous information network, which can integrate various types of infrastructure nodes and their rich relationships (*Gao et al., 2020*). *Zhao, Lang & Liu (2017)* proposed a unified ontology-based model to handle

heterogeneous cyber threat intelligence information. In their model, they mapped threat intelligence from various sources into a unified model, achieving a consistent representation and thereby improving the efficiency of threat intelligence sharing and analysis. They also implemented an intelligent integration framework based on a unified intelligence model and the open-source intelligence collection tool IntelMQ (*Zhao, Lang & Liu, 2017*). *Jo, Lee & Shin (2022)* introduced an automated method for extracting and analyzing network threat intelligence from unstructured text. This method is tailored to the field of network security and includes a neural language model-based Named Entity Recognition (NER) and Relationship Extraction (RE) model (*Jo, Lee & Shin, 2022*). *Al-Hawawreh et al. (2020)* proposed a novel threat intelligence solution based on deep learning technology, designed to discover network threat intelligence from the Space, Air, Ground, and Sea (SAGS) network. This solution comprises three modules: deep pattern extractor, threat intelligence-driven detection, and threat intelligence attack type identification. The deep pattern extractor module aims to uncover hidden patterns in the Internet of Things (IoT) network, with its output serving as input for threat intelligence-driven detection. Threat intelligence attack type identification is used to identify attack types of malicious patterns, helping in response to security incidents (*Al-Hawawreh et al., 2020*).

## Research on entity relationship extraction of network threat intelligence

In terms of extracting security entity relationships, *Perera et al. (2018)* employs natural language processing techniques to automatically classify sentences in input news texts based on the described network attack events. They then complement this with named entity recognition to swiftly detect key elements potentially related to network attacks. Finally, they use rules to extract relationships between these key elements (*Perera et al., 2018*). To facilitate research on security entity relationship extraction, *Phandi, Silva & Lu (2018)* has released a dataset for network security report relationship extraction. This dataset defines a total of four categories of relationships, all of which are defined based on the semantic roles between entities (*Phandi, Silva & Lu, 2018*). *Satyapanich, Finin & Ferraro (2019)* used a deep learning information extraction pipeline to extract network security events and the relationships between them. However, due to the simplicity of the model, it could not capture the interaction features between entities, resulting in poor performance. In addition, *Gasmi, Laval & Bouras (2019)* employed long short-term memory (LSTM) and dependency syntax features to extract relationships between entities. The inclusion of syntax features improved the performance of relationship extraction, indicating that incorporating syntax features can enhance the performance of security entity relationship extraction. The aforementioned works did not consider the connection between entity recognition and relationship extraction tasks, as well as the problem of insufficient labeled corpora for model training (*Gasmi, Laval & Bouras, 2019*). *Zhang et al. (2024)* proposed an end-to-end network security knowledge triple extraction method that integrates adversarial active learning. In the field of network threat intelligence information extraction systems, *Georgescu (2020)* constructed an entity relationship

extraction system using IBM tools on their self-defined ontology model. Unfortunately, this system is not open source (*Georgescu, 2020*). While these methods excel in establishing relationships between entities, they do not effectively handle issues related to relationship overlap and error propagation. *Joshi et al. (2021)* have constructed a series of "semantic triplets" (including subjects, objects, and their mutual relationships) by elaborating on the interactions between cybersecurity entities. They employed graph convolutional neural network technology to analyze and score these triplets. This approach enables researchers to validate and utilize data within cybersecurity knowledge graphs more precisely, thereby conducting research and responding to cybersecurity incidents more efficiently (*Joshi et al., 2021*). *Lin et al. (2024)* and his team proposed a scheme that combines graph convolutional networks (GCN) and graph attention networks (GAT) for detecting ransomware. They used Cuckoo Sandbox to record malicious ransomware behavior and extracted API call sequences from the generated JSON reports for detection (*Li, Qiang & Ma, 2024*). *Li, Qiang & Ma (2024)* designed a method that strengthens cybersecurity through graph neural networks (GNN). Specifically, they integrated cyber threat intelligence data into knowledge graphs and used graph neural networks to conduct in-depth analysis and processing of these graphs (*Lin et al., 2024*).

## METHODS

### Predefined knowledge

**Definition 1** *Threat intelligence entities: Given that cybercriminals frequently exploit online resources for malicious activities, we categorize network threat entities into seven main types: attack activity entities, vulnerability entities, attack object entities, malware entities, tool entities, threat actor entities, and intrusion entities.*

*Threat-actor:* an individual, group, or organization deemed to be malicious.

*Intrusion-set:* an act with the intent to attack.

*Attack campaign:* a clearly defined attack.

*Tool:* This is the software or program code used by the threatened subject to carry out an attack.

*Malwate:* Malware is used for capturing the confidentiality, integrity, or availability of a victim's data or system.

*Vulnerability:* a software error that can be used directly by a hacker to access the system or network.

*Threat-object:* an attacked organization, individual, or object with a specific identity.

**Definition 2** *Threat intelligence entity identification: This refers to the identification of a possible threat entities E, $E_1(w_1, w_2, w_3)$,..., $E_n(w_{n-2}, w_{n-1}, w_n)$ from the sentence $S = (w_1, w_2, w_3, w_4, \ldots w_5)$. E represents the set of entities, $E_1, E_n \in E$, S represents a sentence where w represents the word that constitutes the sentence.*

**Definition 3** *Threat intelligence entity relationship extraction: This refers to identifying all existing entities E from sentence S, and extracting the relationships R between entities,*

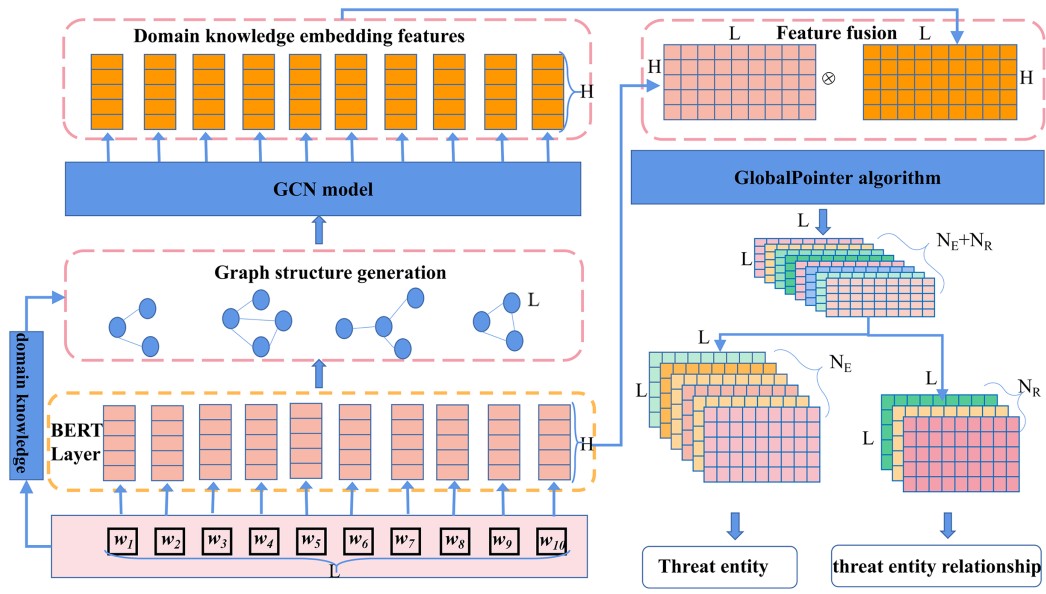

**Figure 3 Threat intelligence entity identification and entity relationship extraction model.**

represented as $R(E_1, E_2)$, where $E_1$ and $E_2$ represent threat entities, and $R(.)$ represents the relationship between entities.

**Definition 4 *Domain knowledge graph:*** *Domain knowledge refers to the collection of knowledge texts in a certain field to form a new set K, and the analysis of the data in set K to obtain the relationships between the words in the domain knowledge text denoted as R. R generates the domain knowledge graph for words in sentence S, represented as $G(S, R)$. G is represented as a graph structure, and S represents the sentence, $S = (w_1, w_2, w_3, w_4, \ldots, w_n)$. R represents the set of relationships between words, $R = (<w_1, w_2>, <w_4, w_5>, \ldots <w_{n-1}, w_n>)$ where $<w_1, w_2>$ indicates a relationship between words $w_1$ and $w_2$.*

## Model overview

To accurately identify threat behavioral entities and extract relationships between them from a vast amount of network threat text, we propose the CtiErRe model. This model introduces a text feature extraction algorithm based on domain knowledge embeddings and incorporates structural information from the text into the features using GCN. Additionally, it employs the BERT model to extract character features from the text. The structural and character feature information is merged using the LayerNorm algorithm. Finally, the GlobalPointer algorithm is employed to predict threat behavioral entities and relationships. The overall mathematical feature representation of the model is shown in Eq. (1), and the structural diagram of the overall model is depicted in Fig. 3.

$$F_{L*n*n} = f_{gp}\big(f_{layernorm}\big(f_{BERT}(X_{b \times n \times 768}) : f_{GCN}(F_{b \times n \times 768}, G_{b_{\times} n \times n})\big)\big) \qquad (1)$$

where $F_{L*n*n}$ represents the model's output features in the form of an $L * n * n$ matrix, where $L$ is the sum of the number of entity types and relationship types, and $n$ is the length of the text. $f_{gp}$, $f_{layernorm}$, $f_{BERT}$, and $f_{GCN}$ represent the GlobalPointer operation, LayerNorm

feature fusion algorithm, BERT model operation, and GCN model operation, respectively. $X_{b \times n \times 768}$ and $F_{b \times n \times 768}$ represent the character feature matrix, where $b$ is the batch size of text, $n$ is the text length, and 768 is the word feature dimension. $G_{b \times n \times n}$ represents the text adjacency matrix based on the domain knowledge graph.

## Domain knowledge embedding

To improve the predictive accuracy of tasks in natural language, it is necessary to enhance the feature representation of text. While using the BERT model for text feature representation has yielded good results in many common NLP tasks, for specific domain NLP tasks, relying solely on features generated by the BERT model for task computation sometimes does not perform very well. To address this issue, this article proposes a domain knowledge embedding algorithm.

The core idea of the domain knowledge embedding algorithm is to collect domain-specific vocabulary, construct a relevance matrix $M$ among these domain knowledge words, and for any sentence $S$, generate a word relevance adjacency matrix $M_L$ within the sentence using the $M$ matrix. The obtained $M_L$ matrix, along with word features $F_W$, is then used for domain knowledge embedding through the GCN network, as represented in Eqs. (2) and (3), which express the mathematical feature representation of domain knowledge embedding.

$$F_R = f_{GCN}(M_L, F_W) \tag{2}$$

$$M_L = \begin{cases} M_{L,i,j} = S_{w_i w_j}, & if \ w_i = w_j \\ M_{L,i,j} = 0, & if \ w_i \neq w_j \end{cases} \tag{3}$$

$F_R$, $F_W$, $w_i$, and $f_{GCN}$ represents the domain knowledge embedding, which serves as the feature output, along with the word feature of the $i$th word in the sentence and the GCN model.

The specific construction method of domain knowledge embedding is as follows:

(1) Collect large amounts of threat intelligence text and mark them by pre-defined entity rules. Collect the tagged entities into a text collection.

(2) The analysis of all the resulting entity text sets gives the word set $W$ often present in the field, and the word frequency matrix $M_P$ is constructed by analyzing the frequency of each word appearing in $W$.

(3) PMI calculations are applied to the word frequency matrix $M_P$, as shown in Eq. (4), to obtain the relationship matrix $M_{PMI}$ between each word and the rest of the words in $W$.

$$PMI(A, B) = \log_2 \frac{P(A, B)}{P(A) \times P(B)} \tag{4}$$

where $A$, $B$ are two words, $P(A, B)$, the frequency of co-occurrence of $A$ and $B$, $P(A)$, the frequency of word $A$, and $P(B)$, the frequency of word $B$.

(4) Using the $M_{PMI}$ matrix, an intra-sentence word-to-word correlation adjacency matrix $M_L$ is generated for threat intelligence information sentence $S$. The values in the $M_{PMI}$ matrix are used for word pairs that exist in it, while 0 is used to represent those that do not.

(5) The matrix $M_L$ and the word features from sentence $S$ are input into the GCN model for computation, resulting in the embedded text features of the final domain knowledge.

## Graph convolutional network

To better integrate the domain knowledge features of the text into the text features, this article uses the domain knowledge text graph to generate a new neighbor matrix with the neighborhood knowledge (*Bhatti et al., 2023*). It integrates the neighborhood knowledge into the text features through GCN (*Phan, Nguyen & Hwang, 2023*).

The essence of the GCN network is to propagate node representations through the adjacency matrix, and the propagation formula is shown in Eq. (5).

$$\mathrm{H}^{k+1} = \sigma\left(\tilde{D}^{-\frac{1}{2}}\widetilde{\mathrm{A}}\tilde{D}^{-\frac{1}{2}}H_k W_k\right) \tag{5}$$

where $k$ represents the number of layers of the convolution. $\delta$ represents an activation function, which can be a ReLU or Tanh function. $H$ represents the characteristics of the node. $A$ represents the adjacency matrix, $\tilde{A} = A + I$, $I$ is a unit matrix. $\tilde{D}$ is a diagonal matrix, $\widetilde{D}_{ii} = \sum_j \tilde{A}_{ij}$. $W$ is a weight parameter that needs to be learned.

For further analysis, the operation of each GCN node to summarize the information from its neighbors is shown in Eq. (6).

$$H_i^k = \sigma\left(\sum_{j\in\{N(i)\cup i\}} \frac{\tilde{A}_{ij}}{\sqrt{\tilde{D}_{ii}\tilde{D}_{jj}}} H_j^{k-1} W^k\right) \tag{6}$$

where $N(i)$ represents the neighbor set of node $i$. GCN updates the new representation calculation and Eq. (7) by combining aggregated information from the neighbor node with the representation from the current node.

$$H_i^k = \sigma\left(\sum_{j\in\{N(i)\}} \frac{A_{ij}}{\sqrt{\tilde{D}_{ii}\tilde{D}_{jj}}} H_j^{k-1} W^k + \frac{1}{\tilde{D}_i} H_i^{k-1} W^k\right). \tag{7}$$

## BERT

Bidirectional encoder representations from transformers (BERT) is a pre-trained language model based on the Transformer architecture, designed to understand the semantics of language through deep bidirectional context. Unlike traditional unidirectional language models, BERT is trained by considering both the preceding and following context in the sentence, enabling it to capture word meanings and sentence structures more accurately.

---

**Algorithm 1  The LayerNorm feature fusion.**

**Input:** The features output by the BERT model are denoted as $X_b$, and those by the HAN model are denoted as $X_g$

**Output:** Fused features $X_F$

1:  $X_{new\_g} \Leftarrow$ Reshape$(X_g, X_b)$ #The tensor shape of $X_g$ is unified with the tensor shape of $X_b$

2:  $\beta \Leftarrow X_{new\_g} {}^*W_1$ #Take a linear transformation of $X_{new\_g}$ with $W_1$ as the argument

3:  $\gamma \Leftarrow X_{new\_g} {}^*W_2$ #Take a linear transformation of $X_{new\_g}$ with $W_2$ as the argument

4:  $\mu \Leftarrow$ Mean$(X_b)$ #Calculate the mean of $X_b$

5:  $\sigma \Leftarrow$ Variance$(X_b)$ #Calculate the variance of $X_b$

6:  $X_{new\_b} \Leftarrow (X_b - \mu)$ / sqrt$(\sigma)$ #Recalculate $X_b$ using the normal distribution

7:  $X_F \Leftarrow \gamma {}^*X_{new\_b} + \beta$ #Two tensors are fused

8:  **return** $X_F$

---

The BERT model employs the encoder component of a multilayer Transformer to learn general knowledge through pretraining tasks, which is then transferred to perform downstream tasks. The BERT architecture is composed of multiple layers of embedding (*Koroteev, 2021*). Specifically, the embedding layer in BERT consists of three components: token embeddings, segment embeddings, and position embeddings. The token embedding layer is a standard embedding layer. The segment embedding layer is used to handle the sentence pair classification task, while the position embedding layer encodes the positions of words within a sentence. In summary, the BERT model integrates multiple embedding layers with attention mechanisms. Together, the embedding layers and attention mechanisms form the Transformer model, with BERT consisting of multiple Transformer layers (*Jawahar, Sagot & Seddah, 2019*).

## Feature fusion

To well integrate the text features obtained based on the BERT model and the text feature information obtained based on the GCN, we carried out the feature fusion method. The fusion method is shown in Eq. (8).

$$\mathrm{F} = f_{\mathrm{MLP}}\big(\mathrm{layernorm}\big(\mathrm{cat}\big(X_b : X_g\big)\big)\big) \tag{8}$$

where *cat(:)* indicates that the two matrices are joined according to the last dimension. $X_b$ represents the text features of BERT model output. $X_g$ represents the textual features of the GCN model output. The LayerNorm algorithm is a feature fusion technique that transforms two input tensors into features with identical dimensionality, which is conducive to the execution of downstream tasks. Among them, *layernorm(.)* The calculation method is shown in Algorithm 1. $f_{MLP}$ is a fully connected neural network.

## GlobalPointer algorithm

GlobalPointer uses a global normalization approach to predict tasks. This algorithm class is similar to a multi-head attention mechanism, the number of heads is the number of labels in the task. The core idea of the method is as follows: First, expand the input feature tensor $X \in R^{L*d}$ to $X_{new} \in R^{L*\mathbf{d}*n}$; then perform a linear transformation on $X_{new}$ to obtain

---

**Algorithm 2  GlobalPointer algorithm.**

---

**Input:** Attention mechanism head number, *heads*, the size of each head, *head_size*, and the input data, *inputs*.

**Output:** (*inputs.shape*[0],*heads*,

*inputs.shape*[1],*inputs.shape*[1])type of tensor

1:  *inputs* ⟸ dense*(inputs)* #The dense is a Dense operation

2:  *inputs* ⟸ split(*inputs*, *self.heads*, *axis* = −1) #split is a segmentation function

3:  *inputs* ⟸ Keras.stack(*inputs*, *axis* = −2) #Merge the tensors in the inputs list in the penultimate dimension.

4:  *qw* ⟸ *inputs*[...,: *head_size*]

5:  *kw* ⟸ *inputs*[..., *head_size* :]

6:  *qw, kw* ⟸ RoPE(*qw,kw*) #RoPE rotary encoding

7:  *logits* ⟸ *qw* × *kw* #Calculate the internal product

8:  *logits* ⟸ sequence_masking(*logits,mask*) #exclude the padding mask as a mask

9:  *mask* ⟸ Compute the lower trigonometric matrix of *logits*

10:  *logits* ⟸ *logits* − (1 − *mask*) ∗ $e^{12}$

11:  Return *logits* ⟸ *logits* / *self.head_size* ∗ ∗0.5

---

the $Q$ and $K$ tensors, and use the rotation-based encoding to regenerate $Q_{RoPE}$ and $K_{RoPE}$; finally, calculate the inner product of $Q_{RoPE}$ and $K_{RoPE}$ to obtain the $G \in R^{L*L*n}$ tensor. Equations (9) to (12) represent the mathematical expression of the entire algorithm.

$$X_{new} = \exp end\_\dim(X, n), X \in R^{L*d}, X_{new} \in R^{L*\mathbf{d}*n} \tag{9}$$

$$Q_{RoPE} = \Re * W_1 * X_{new} \tag{10}$$

$$K_{RoPE} = \Re * W_2 * X_{new} \tag{11}$$

$$G = Q_{EoPE}^{\mathrm{T}} * K_{EoPE} = X_{new}^T * W_1^{\mathrm{T}} * \Re^T * \Re * W_2 * W_{new} \tag{12}$$

where *expend_dim(.)* represents dimension expansion of tensors; $W_1$ and $W_2$ represent the parameters of linear transformation; *RoPE* stands for Rotation-based Positional Encoding; $\Re$ is a rotation encoding matrix. The specific approach of the GlobalPointer algorithm is presented in Algorithm 2.

The GlobalPointer algorithm transforms traditional sequence prediction into a matrix coordinate format. This format is no longer limited to predicting labels for individual words but extends to predicting labels for spans of text. As shown in Fig. 4, the threat entities in the sentence are converted into a matrix, where the red positions in the matrix represent the coordinates of the threat entities within the sentence. For example, the "Threat Actor" entity in the sentence has red position coordinates $(1, 2)$, indicating that the text from position 1 to position 2 corresponds to the "Threat Actor" entity (with the first position starting at 0).

To further tailor the model for the threat entity relation extraction task, we improved the representation of relationships between threat entities. We adopted a full-combination

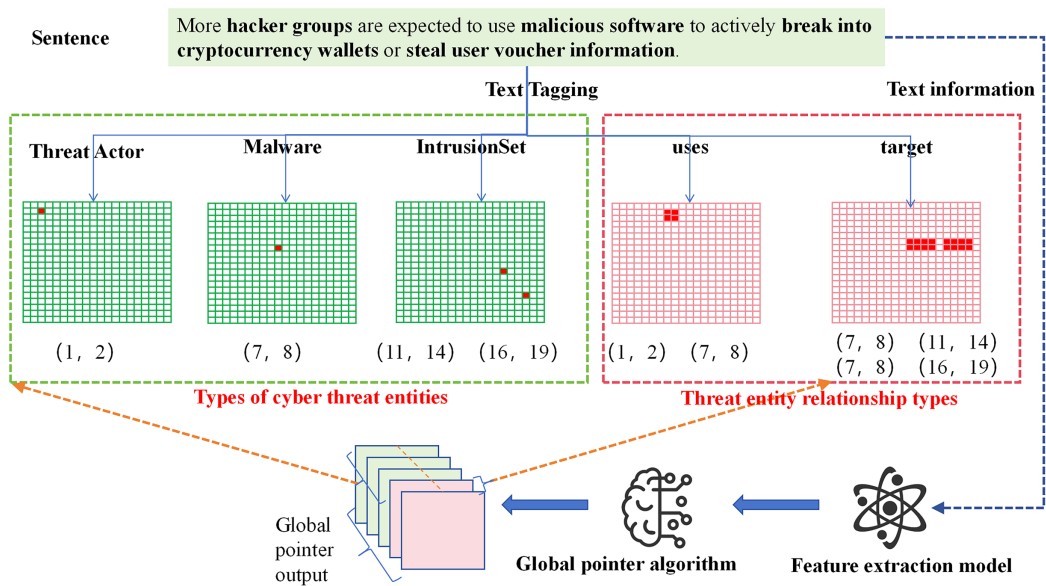

**Figure 4** Flowchart of network threat entity and relationship extraction based on GlobalPointer.

approach between threat entities, combining corresponding words between entities, and marked the positions in the matrix accordingly. For instance, the "uses" relation between the threat entity at position (1, 2) and the threat entity at position (7, 8) is represented by the full combination of positions, with the corresponding positions (1, 7), (1, 8), (2, 7), and (2, 8) marked in red. Through these operations, we can simultaneously extract both the threat entities and the relationships between them in the sentence.

# EXPERIMENT AND DISCUSSION

## Data set

To verify the effectiveness of our proposed model, the data provided in the literature was utilized for validation (https://github.com/MuYu-z/CDTier; *Zhou et al., 2023*). Modifications were made to the labels, resulting in a total of 9,549 entity tags. Following the STIX format, this article defined seven network threat entities, namely: threat-actor (TA), intrusion-set (IS), attack campaign (AC), tool (TO), malware (MW), vulnerability (VUL), and threat-object (TOB). The specific distribution of each entity is as follows: there are 1,353 campaign entities, 2,052 vulnerability entities, 1,486 threat-object entities, 1,100 malware entities, 906 tool entities, 1,094 threat-actor entities, and 1,558 intrusion-set entities. Figure 5 shows the distribution of each entity intuitively. At the same time, we annotated the relationships of entities, totaling 2,193 relational data entries. These entities communicate with each other through both uses and targets. In this article, we identified a total of 8,038 relationships, with 3,396 classified as "uses" and 4,642 as "Targets". As not all of the seven entities interact with each other, we have re-defined 20 specific relationship triples of threat entities by the STIX framework. The specific relationship triples are shown in Table 1. Figure 6 is the distribution of relational triples, where 1,2,…,20, indicates the serial number of each triplet (https://github.com/RENWENO/CTIER).
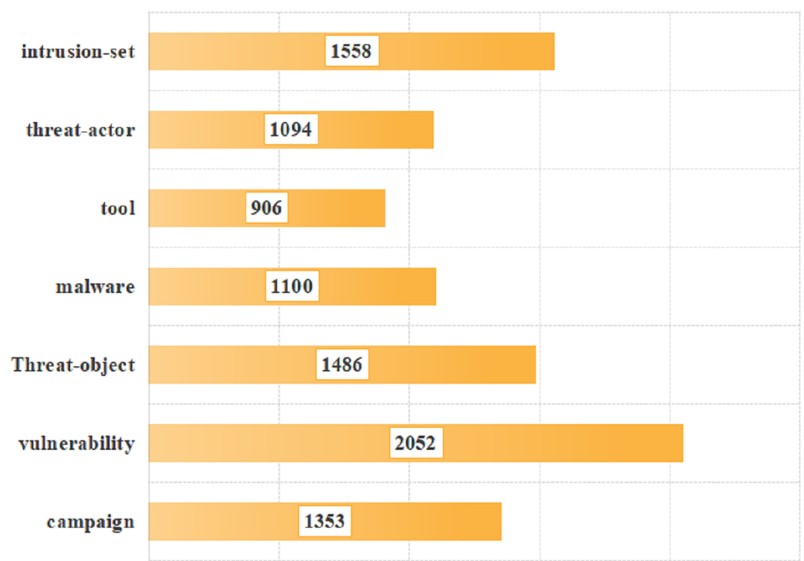

**Figure 5 Distribution of threat entities in the dataset.**

**Table 1 Describes the relational triples of 20 threat entities.**

| Number | Triplet type | Description |
|---|---|---|
| 1 | TA uses AC | Indicates that there is a use relationship between the threat actor (TA) and the attack campaign (AC). |
| 2 | TA uses TO | Indicates that there is a use relationship between the threat actor (TA) and the tool (TO). |
| 3 | TA uses VUL | Indicates that there is a use relationship between a threat actor (TA) and a vulnerability (VUL). |
| 4 | TA uses MW | Indicates that there is a uses relationship between a threat actor (TA) and malware (MW). |
| 5 | TA uses TOB | Indicates that there is a use relationship between the threat actor (TA) and the threat object (TOB). |
| 6 | TA uses IS | Indicates that there IS a use relationship between a threat actor (TA) and an intrusion set (IS). |
| 7 | IS uses AC | Indicates that there IS a use relationship between the intrusion set (IS) and the attack campaign (AC). |
| 8 | AC uses TO | Indicates that there is a relationship between the attack campaign (AC) and the tool (TO). |
| 9 | AC uses MW | Indicates that there is a use relationship between the attack campaign (AC) and the malware (MW). |
| 10 | IS uses TO | Indicates that there IS a relationship between the intrusion set (IS) and the tool (TO). |
| 11 | IS uses VUL | Indicates that there IS a use relationship between an intrusion set (IS) and a vulnerability (VUL). |
| 12 | IS uses MW | Indicates that there IS a use relationship between the intrusion set (IS) and malware (MW). |
| 13 | MW uses VUL | Indicates a use relationship between malware (MW) and vulnerability (VUL). |
| 14 | AC targets VUL | Indicates that there is a target relationship between attack campaign (AC) and vulnerability (VUL). |
| 15 | AC targets TOB | Indicates that there is a target relationship between attack campaign (AC) and threat object (TOB). |
| 16 | TO targets VUL | Indicates that there is a target relationship between the tool (TO) and the vulnerability (VUL). |
| 17 | MW targets TOB | Indicates that there is a target relationship between malware (MW) and threat object (TOB). |
| 18 | TA targets TOB | Indicates that there is a target relationship between threat subject (TA) and threat object (TOB). |
| 19 | IS targets TOB | Indicates that there is a target relationship between intrusion set (IS) and threat object (TOB). |
| 20 | VUL targets TOB | Indicates that there is a target relationship between vulnerability (VUL) and threat object (TOB). |

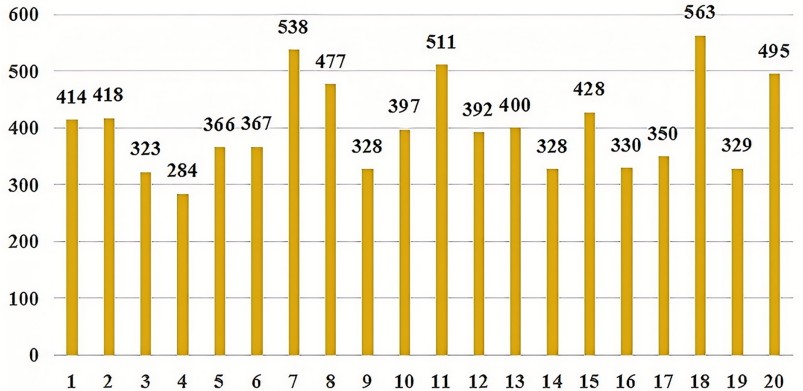

**Figure 6 Quantity distribution of 20 threat entity relationship triples.**

## Metrics

The evaluation metrics that we used in this study mainly include precision (P) (Eq. 13), recall (R) (Eq. 14), F1 (Eq. 15), and accuracy (acc) (Eq. 16).

$$\textbf{Precision} = \frac{TP}{TP + FP} \tag{13}$$

$$\textbf{Recall} = \frac{TP}{TP + FN} \tag{14}$$

$$\textbf{F1} = \frac{2 \times \text{Precision} \times \text{Recall}}{\textit{Precision} + \text{Recall}} \tag{15}$$

$$\textbf{acc} = \frac{TP + \text{TN}}{TP + \text{TN} + FP + \text{FN}} \tag{16}$$

## Baselines

**BI LSTM+CRF (BC) model:** This model is a deep learning approach that combines bidirectional long short-term memory (Bi-LSTM) networks with conditional random fields (CRF). It is primarily used for sequence labeling tasks, particularly excelling in NER tasks in NLP (*Gasmi, Bouras & Laval, 2018*).

BERT+BI LSTM+CRF (BBC) model: This model combines the contextual understanding capability of BERT, the bidirectional information capture ability of Bi-LSTM, and the sequence labeling optimization power of CRF. It leverages BERT's deep semantic understanding, Bi-LSTM's ability to capture long- and short-term dependencies, and CRF's ability to predict globally optimal label sequences. Compared to single models, the BBC model demonstrates higher accuracy in experiments (*Li et al., 2022*).

ABERT+BI LSTM+CRF (ABC) model: This model integrates the lightweight characteristics of ALBERT, the bidirectional contextual capturing ability of Bi-LSTM, and the sequence labeling optimization capability of CRF. It is particularly suited for NER tasks, providing precise entity boundaries and category labels. ALBERT reduces the number of model parameters while maintaining BERT's performance and improving

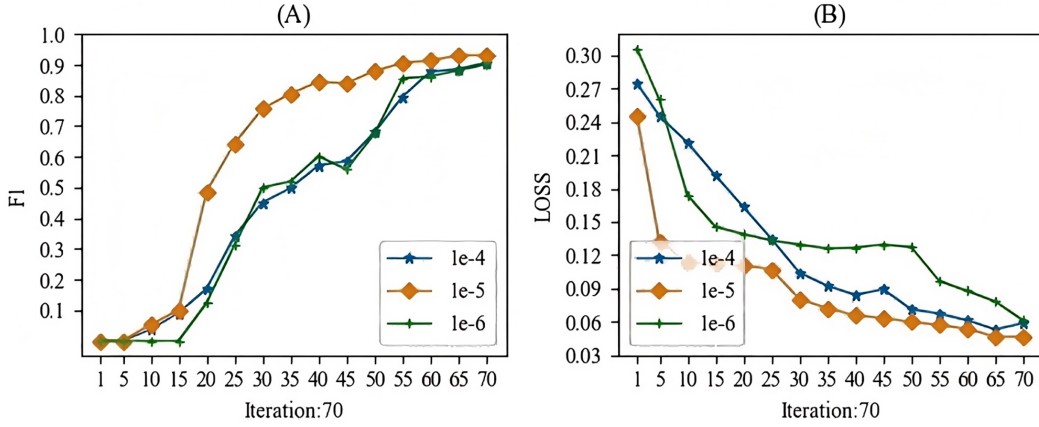

**Figure 7 The learning rate parameter experiment for the entity recognition task.**

efficiency; Bi-LSTM handles long- and short-term dependencies; CRF ensures globally optimal label sequences (*Ren et al., 2022*).

**CasRel model:** This model is an innovative framework for relation extraction, designed to extract relational triples from unstructured text. The core idea of the model is to treat relations as functions that map the subject to the object, rather than treating relations as discrete labels (*Wei et al., 2019*).

**AGGCN model:** This model distinguishes the correlations between nodes and edges by converting the dependency tree into a weighted graph, thereby capturing the dependencies within a sentence more effectively. Unlike fully connected graphs, AGGCNs preserve the structure of the original dependency tree and dynamically adjust the importance of nodes and edges through an attention mechanism, enabling more precise relation extraction (*Guo, Zhang & Lu, 2019*).

## Hyperparameter selection

When training the models mentioned in this article, the main hyperparameter we focused on was the learning rate. In this study, we experimented with learning rates ranging from 1e-1 to 1e-10. During the experimental process, we observed that when the learning rate was higher than 1e-3, gradient vanishing occurred, and when the learning rate was lower than 1e-7, the gradient also vanished. Therefore, we selected a learning rate between 1e-4 and 1e-6 for this study. The batch training results for the specific models are shown in Figs. 7 and 8. From these Figures, we can see that our choice of a learning rate at 1e-5 yielded good results.

### *Effect of domain knowledge embedding*

This article proposes a domain knowledge embedding method for threat entity recognition and relationship extraction. To validate the effectiveness of domain knowledge embedding, we conducted a set of comparative experiments, testing models with and without domain knowledge on threat intelligence entity recognition and relationship extraction tasks. From Table 2, it can be observed that the domain knowledge embeddings we extracted have a

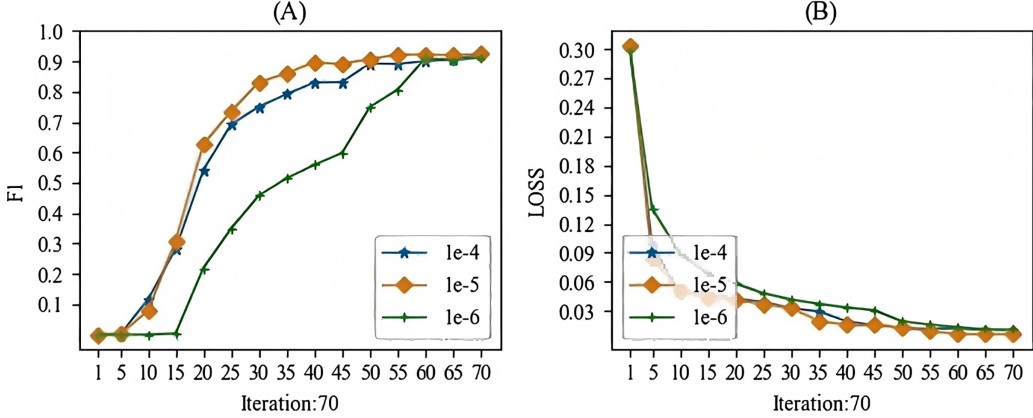

**Figure 8 The learning rate parameter experiment for the relation extraction task.**

**Table 2 Effect of domain knowledge embedding.**

|  | Add domain knowledge embedding | No domain knowledge embedding |
| --- | --- | --- |
| Entity recognition task (ER) (F1) | 93.11 | 91.02 |
| Relation extraction task (RE) (F1) | 92.45 | 90.23 |
| Entity recognition task (ER) (Precision) | 92.78 | 90.72 |
| Relation extraction task (RE) (Precision) | 92.34 | 90.16 |
| Entity recognition task (ER) (Recall) | 93.44 | 91.32 |
| Relation extraction task (RE) (Recall) | 92.56 | 90.30 |

**Table 3 Results of our proposed methods.**

|  | Our model |
| --- | --- |
| Entity recognition task (ER) (acc) | 92.13 |
| Relation extraction task (RE) (acc) | 91.45 |
| Entity recognition task (ER) (F1) | 93.11 |
| Relation extraction task (RE) (F1) | 92.45 |
| Entity recognition task (ER) (Precision) | 92.78 |
| Relation extraction task (RE) (Precision) | 92.34 |
| Entity recognition task (ER) (Recall) | 93.44 |
| Relation extraction task (RE) (Recall) | 92.56 |

significant impact on threat intelligence entity recognition and relationship extraction tasks. The inclusion of domain knowledge embeddings improved the model's recognition and extraction tasks by 2%.

## Overall effect of the model

In this section, the impact of each metric of our proposed model on two tasks will be detailed. It can be seen from Table 3 that in the threat entity recognition task, the values of

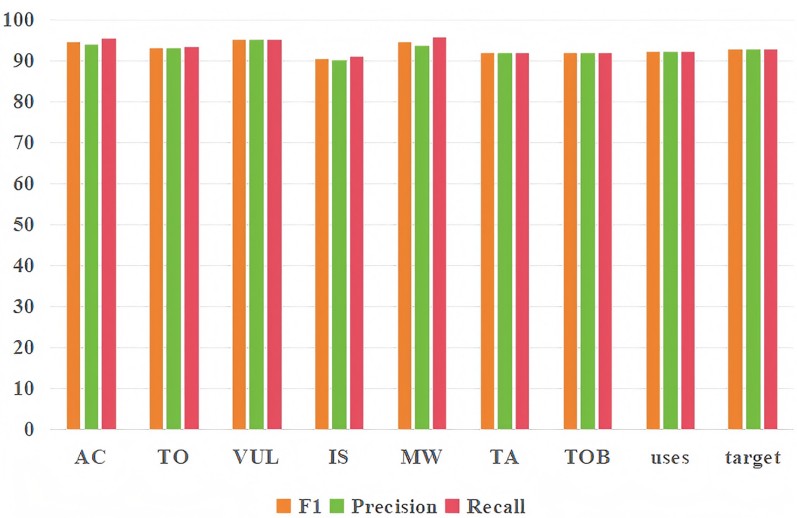

**Figure 9 Entity recognition and relationship extraction.** Each type of entity and relationship prediction effect.

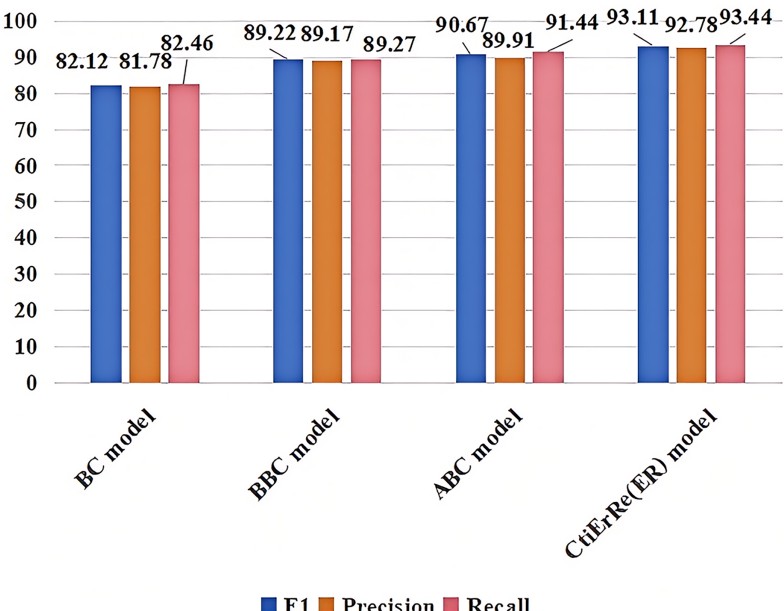

**Figure 10 Comparison of threat entity recognition models.**

acc, precision, recall and F1 of our proposed model can reach 92.13%, 92.78%, 93.44% and 93.11% respectively. In relation extraction task, the values of acc, precision, recall and F1 of the model proposed by us can reach 91.45%, 92.34%, 92.56% and 92.45%. At the same time, we found that the index of the entity recognition task is higher than that of the relationship extraction task, mainly because the index of predicting the occurrence of both entity and relationship in the relationship extraction task is lower. Figure 9 shows the predictive effects of seven threat entities and two relationships. It can be seen from the

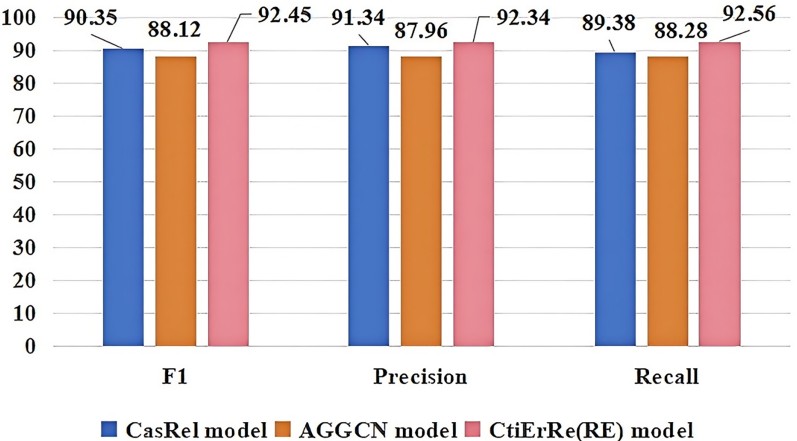

**Figure 11 Comparison of threat entity relationship extraction models.**

figure that precision, recall and F1 can reach more than 90% in the test of each entity type and relationship type.

## Comparison with other models

The models compared with the threat entity recognition model are BI_LSTM+CRF (BC) (*Gasmi, Bouras & Laval, 2018*), BERT+BI_LSTM+CRF (BBC) (*Li et al., 2022*), ABERT +BI_LSTM+CRF (ABC) (*Ren et al., 2022*). The models compared with the threat entity relationship extraction model are CasRel model (*Wei et al., 2019*), AGGCN model (*Guo, Zhang & Lu, 2019*). Figure 10 compares our threat entity recognition model with other models in terms of F1, precision, and recall metrics. It can be observed that our model outperforms the others by 9% to 2% in these three metrics. Figure 11 compares our threat entity relation extraction model with existing relation extraction models. We can observe that our proposed threat entity relation extraction model outperforms existing models by approximately 4% in terms of F1, precision, and recall. Table 4 displays the recognition performance of various entities in the threat entity recognition model. Table 5 showcases the extraction performance of various relationships in the threat entity relation extraction model.

## DISCUSSION

### Overall model analysis

This article proposes a threat intelligence entity recognition and relationship extraction model based on domain knowledge embedding (CtiErRe). The model has a total parameter count of 125,416,704 and the trained model size is 280 MB. The proposed model is a combination of the BERT model, GCN model and GlobalPointer algorithm. Among them, the complexity of the BERT model is $O(N^2)$ (*Eltayeb, 2024*), the time complexity of the GCN model is $O(|E| * K)$, where $|E|$ represents the number of non-zero elements in the sparse matrix indicating the number of edges in the graph, and K

**Table 4 The identification effect of each threat entity in each model.**

|  |  | BC model | BBC model | ABC model | CtiERRE model |
|---|---|---|---|---|---|
| AC | F1 | 82.42 | 89.34 | 91.65 | 94.54 |
|  | Precision | 82.77 | 88.83 | 90.19 | 93.8 |
|  | Recall | 82.07 | 89.86 | 93.16 | 95.29 |
| TO | F1 | 81.57 | 89.01 | 90.87 | 93.13 |
|  | Precision | 80.85 | 89.37 | 89.28 | 93.01 |
|  | Recall | 82.3 | 88.65 | 92.52 | 93.25 |
| VUL | F1 | 84.78 | 90.43 | 90.42 | 95.19 |
|  | Precision | 82.72 | 90.48 | 90.49 | 95.25 |
|  | Recall | 86.95 | 90.38 | 90.35 | 95.13 |
| IS | F1 | 81.42 | 88.99 | 90.98 | 90.52 |
|  | Precision | 81.04 | 88.62 | 90.33 | 89.99 |
|  | Recall | 81.8 | 89.36 | 91.64 | 91.06 |
| MW | F1 | 81.22 | 88.34 | 90.18 | 94.65 |
|  | Precision | 81.55 | 88.33 | 89.38 | 93.65 |
|  | Recall | 80.89 | 88.35 | 90.99 | 95.67 |
| TA | F1 | 81.79 | 88.97 | 89.95 | 91.88 |
|  | Precision | 81.77 | 88.95 | 89.82 | 91.74 |
|  | Recall | 81.81 | 88.99 | 90.08 | 92.02 |
| TOB | F1 | 81.66 | 89.45 | 90.64 | 91.88 |
|  | Precision | 81.73 | 89.60 | 89.92 | 91.99 |
|  | Recall | 81.59 | 89.30 | 91.37 | 91.77 |

**Table 5 Effect of entity relationship extraction for each threat model.**

|  | CasRel model | | | AGGCN model | | | CtiErRe(RE) model | | |
|---|---|---|---|---|---|---|---|---|---|
|  | F1 | Precision | Recall | F1 | Precision | Recall | F1 | Precision | Recall |
| Uses | 90.28 | 91.81 | 88.8 | 88.52 | 88.94 | 88.1 | 92.15 | 92.04 | 92.26 |
| Target | 90.42 | 90.86 | 89.98 | 87.72 | 86.98 | 88.47 | 92.74 | 92.63 | 92.85 |

represents the number of layers in the GCN model. The time complexity of the GlobalPointer algorithm is $O(N^2)$. Therefore, the overall time complexity of the CtiErRe model proposed by us is $O(N^2) + O(|E| * K) + O(N^2)$.

## Case analysis

The CtiErRe model can identify both threat entities and the relationships between them in threat intelligence. To illustrate its accuracy, consider the following two sentences as examples: "More hacker groups are expected to use malicious software to actively break into cryptocurrency wallets or steal user voucher information." The threat intelligence entities present in the sentence are as follows: Threat actor: threat actor entity: "hacker

**Table 6  The CtiErRe model predicted result for sentence.**

| | |
|---|---|
| input1 | **More hacker groups are expected to use malicious software to actively break into cryptocurrency wallets or steal user voucher information.** |
| *Threat intelligence entity (output)* | **Threat Actor:** hacker groups; **Malware:** malicious software; **Intrusion Set:** break into cryptocurrency wallets; Intrusion Set: steal user voucher information |
| *Threatening entity relationship (output)* | Hacker groups **uses** malicious ware; malicious ware **target** break into cryptocurrency; malicious ware **target** steal user voucher information |
| input2 | **Valery Marchive of LegMagIT found samples of the REvil ransomware used in the Acer attack; At the same time, BleepingComputer also found samples, and from the ransom note, the contents of the conversation between the victim and the attacker, further confirmed the fact that Acer was hit by the REvil ransomware attack and was demanded $50 million ransom.** |
| *Threat intelligence entity (output)* | **Attack campaign:** Acer attack; **Malwate:** REvil ransomware; Intrusion-set: ransom note; **Intrusion-set:** demanded $50 million ransom; **Threat-object:** Acer; **Malwate:** REvil ransomware; **Threat-actor:** REvil |
| *Threatening entity relationship (output)* | Acer attack **uses** REvil ransomware; Acer attack **uses** ransom note; Acer attack **target** demanded $50 million ransom; REvil **uses** REvil ransomware; REvil **target** Acer |

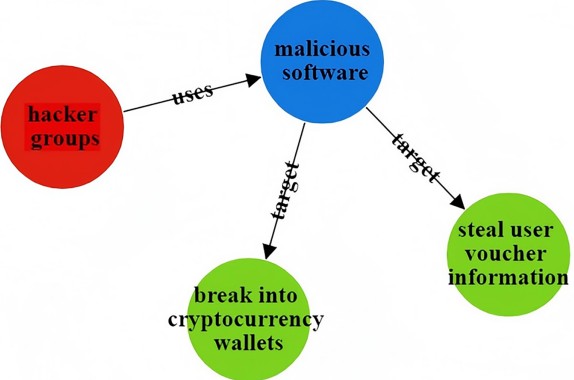

**Figure 12  Sentence 1 result visualization.**

groups", malware entity: "malicious ware", intrusion set entity: "break into cryptocurrency wallets", intrusion set entity: "steal user voucher information". The relationships between threat entities include "uses" and "targets". "Valery Marchive of LegMagIT found samples of the REvil ransomware used in the Acer attack; At the same time, BleepingComputer also found samples, and from the ransom note, the contents of the conversation between the victim and the attacker, further confirmed the fact that Acer was hit by the REvil ransomware attack and was demanded $50 million ransom". The threat intelligence entities present in the sentence are as follows: Attack campaign: "Acer attack", Malwate: "REvil ransomware", Intrusion-set: "ransom note", Intrusion-set: "demanded $50 million ransom", Threat-object: "Acer", Malwate: "REvil ransomware", Threat-actor: "REvil". The relationship between threat entities also includes "use" and "target". Table 6 provides specific output details. Table 6 provides specific output details. Figures 12 and 13 visualizes the results of threat entity recognition and entity relationship extraction for two sentences. In the figure, blue represents malware, red represents attacking entities, green

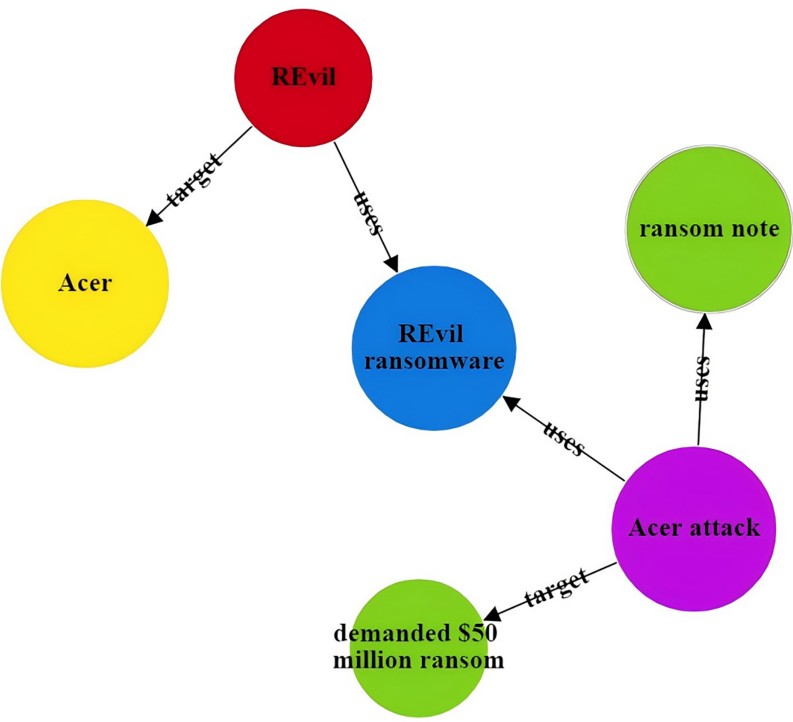

**Figure 13 Sentence 2 result visualization.**

represents intrusion sets, purple represents attack activities, and yellow represents attack objects.

## CONCLUSION

This article proposes a network threat intelligence entity and relationship extraction method that combines domain knowledge embedding with learning neural networks. This method utilizes domain knowledge and graph neural networks for feature extraction and employs the GlobalPointer algorithm for relationship extraction. Specifically, the CtiErRe model utilizes GCN networks to incorporate context relevance and threat entity text relevance features into text features. It employs the BERT model to extract text features and utilizes the GlobalPointer algorithm for the final entity recognition and relationship identification, thereby avoiding error propagation and enhancing the accuracy of overlapping relationship recognition. Additionally, to make threat entities more comprehensive, we have defined seven types of threat entities, namely, threat actor, intrusion set, campaign, tool, malware, vulnerability, and threat object. We have also defined seven relationships between entities, which include "uses" and "targets." Furthermore, we have validated the proposed model, achieving the following metrics in the threat entity recognition task: an accuracy (acc) of 92.13%, precision of 92.78%, recall of 93.44%, and an F1 score of 93.11%. For the relationship extraction task, our model achieved an accuracy (acc) of 91.45%, precision of 92.34%, recall of 92.56%, and an F1 score of 92.45%. While our proposed CtiErRe model has shown promising results in

predicting threat intelligence entities and entity relationships, it is important to note that our training dataset is not extensive enough. To enhance the model's robustness, we plan to augment our dataset with additional data in future work. Additionally, our algorithm for domain knowledge embedding based on graph neural networks is relatively simplistic. In the future, we will explore various graph network models to better encode node representations and extend the representation fusion strategy to a wider range of tasks.

### Funding

This work was supported by the National Natural Science Foundation of China (62262013), the Youth Science and Technology Talent Growth Project of Guizhou Provincial Education Department (No. QJJ2024274), the Education Department of Hainan Province, project number: Hnky2024-79, the Industry-University-Research Innovation Fund for Chinese Universities (Grant No. 2022HS068). The funders had no role in study design, data collection and analysis, decision to publish, or preparation of the manuscript.

### Grant Disclosures

The following grant information was disclosed by the authors:
National Natural Science Foundation of China: 62262013.
The Youth Science and Technology Talent Growth Project of Guizhou Provincial Education Department: QJJ2024274.
Education Department of Hainan Province: Hnky2024-79.
Industry-University-Research Innovation Fund for Chinese Universities: 2022HS068.

### Competing Interests

The authors declare that they have no competing interests.

### Author Contributions

- Gan Liu conceived and designed the experiments, performed the experiments, analyzed the data, performed the computation work, prepared figures and/or tables, authored or reviewed drafts of the article, and approved the final draft.
- Kai Lu conceived and designed the experiments, performed the experiments, analyzed the data, performed the computation work, authored or reviewed drafts of the article, and approved the final draft.
- Saiqi Pi conceived and designed the experiments, performed the experiments, analyzed the data, authored or reviewed drafts of the article, and approved the final draft.

### Data Availability

The data is available at GitHub: https://github.com/RENWENO/CTIER.
The code is available at GitHub: https://github.com/RENWENO/CTIERRE/tree/main.

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
