# Peer review of "Graph neural networks embedded with domain knowledge for cyber threat intelligence entity and relationship mining"

_PeerJ Computer Science, doi:10.7717/peerj-cs.2769_

## Round 0.1 · original submission · Major Revisions

Dear Authors,

Your article has been reviewed. Based on reviewers' opinions, it needs major revisions before being accepted for publication in PeerJ Computer Science.
More precisely, the following points need to be addressed and resolved in the revised version of the manuscript:

1) The novelty of the proposed approach (the Globalpointer algorithm) must be better clarified because it combines existing methods. Also, the statement "Finally, the Globalpointer algorithm is employed to recognize relationships between threat entities and entities" must be better clarified to explain the method's performance.

2) The different metrics reported in Table 4 must be comparable.

3) Although the experimental design appears to be well-planned, with a sufficient number of experiments performed to validate the proposed model, it would be useful to include additional details regarding the choice of baseline models for comparison. The authors' rationale for selecting these baselines should be clearly stated, and the paper should clarify how these models represent the state-of-the-art.

4) All typos must be fixed.

Reviewer 1 ·

Basic reporting

1. Authors should carefully check for grammatical errors and typos.
2. Authors should provide a detailed description about how to extract the relationship from texts.
3. Authors should further highlight their motivations and contributions.

Experimental design

no comment

Validity of the findings

no comment

Additional comments

no comment

Reviewer 2 ·

Basic reporting

The article is generally well-written, but there are some areas where the clarity of expression could be improved. For example, certain technical terms, such as BERT, are introduced without sufficient explanation for non-specialists.

The literature review provides a good contextual background, but it would be strengthened by including more recent references on Graph Neural Networks (GCN) and their application to natural language processing and cybersecurity.

The article is well-structured, with a logical flow from the introduction to the methodology, experiments, and results. However, some figures’ resolutions could be improved. For example, Figures 6 and 7. Some figures are the reformatting of existing works and could be deleted. For example, figures 4 and 5.

While the paper discusses extensive experimentation, it is unclear whether the raw data used in the experiments is publicly available. Providing a link to the modified dataset or indicating if it is proprietary will improve the transparency and reproducibility of the study.

The paper is generally self-contained and well-linked to the hypotheses.

Experimental design

The experimental design appears to be well-planned, with a sufficient number of experiments performed to validate the proposed model. However, it would be useful to include additional details regarding the choice of baseline models for comparison. The rationale for selecting these baselines should be clearly stated, and the paper should clarify how these models are representative of the state-of-the-art.

For the ablation study part, the authors actually did a hyperparameter tuning instead of an ablation study to investigate the effect of different components of the proposed framework.

Validity of the findings

The authors have not explicitly mentioned whether the underlying data and code are made publicly available for replication purposes. To ensure transparency and encourage meaningful replication, it is recommended that the authors share the dataset and model code in an open repository.

Additional comments

None

Reviewer 3 ·

Basic reporting

The English used in this paper is both professional and fluent, and extensive research has been conducted on related work. The structure of the article is well-organized, and the experimental results are detailed and consistent with the hypotheses. However, the first challenge it faces is questionable, as there are some flaws in the chart drawing and some formulas in the formal expression are not standardized enough.

Experimental design

The methods detailed in this paper provide sufficient information to replicate the experiments. However, the research questions are not appropriate. Regarding challenge (1), the author argues that current Named Entity Recognition (NER) methods overlook contextual information, whereas BERT, a well-established approach, takes such information into account and is considered a classic method for NER. For challenge (3), the Globalpointer algorithm is a one-stage method that can address these issues.

Validity of the findings

The experimental method applied in this paper is appropriate, the data presented is detailed, and it effectively addresses the problems raised by the author.

Additional comments

The issues that need to be addressed include:
(1)The proposed approach in the paper appears to be a combination of existing methods. For challenges 1 and 3, there are already established methods to address them, thus making the novelty ambiguous.
(2)In the abstract, the authors mention “Finally, the Globalpointer algorithm is employed to recognize relationships between threat entities and entities.” It’s unclear whether the relationship is identified only between threat and benign entities, or between all entities.
(3)In Table 4, the bold formatting should indicate the highest value for the same metric across different models for a specific category of entities. However, the bold formatting only highlights the highest value in this column, which includes multiple metrics and may make the different metrics non-comparable.
(4)There are some typos and unclear points. For instance, in Equation (8), should “f_MIP” be ”f_MLP”? In line 334 of Algorithm 1, is it correct that X_{new_b}<=(X_b-μ)/sqr(σ)? In line 359 of Algorithm2, what’s the meaning of axis = -2? In the 357 of Algorithm2, “ operatio”->”operation”? Additionally, there misses an equation operation in Equation (13).
(5)Some redundant abbreviations, such as Cyber Threat Intelligence (CTI) in the Introduction section.
(6)Some figures are too large and should be appropriately adjusted, such as Fig. 2, Fig. 10-12.

---

## Round 0.2 · accepted · Accept

Dear Authors,
Your paper has been revised. It has been accepted fo publication in PEERJ Computer Science. Thank you for your fine contribution.

Reviewer 1 ·

Basic reporting

The authors have addressed all my comments, it has been accepted for publication.

Experimental design

The authors have addressed all my comments, it has been accepted for publication.

Validity of the findings

The authors have addressed all my comments, it has been accepted for publication.

Additional comments

The authors have addressed all my comments, it has been accepted for publication.